# OV16 Seroprevalence among Persons with Epilepsy in Onchocerciasis Endemic Regions: A Multi-Country Study

**DOI:** 10.3390/pathogens9100847

**Published:** 2020-10-16

**Authors:** Alfred Dusabimana, Dan Bhwana, Michel Mandro, Bruno P. Mmbando, Joseph N. Siewe Fodjo, Robert Colebunders

**Affiliations:** 1Global Health Institute, Gouverneur Kinsbergen Centrum, University of Antwerp, Doornstraat 331, 2610 Antwerp, Belgium; alfred.dusabimana@uantwerpen.be (A.D.); josephnelson.siewefodjo@uantwerpen.be (J.N.S.F.); 2National Institute for Medical Research, Tanga Centre, P.O. Box 5004, Tanga, Tanzania; dan.bhwana@nimr.or.tz (D.B.); bruno.mmbando@nimr.or.tz (B.P.M.); 3Provincial Health Division Ituri, Ministry of Health, Bunia, P.O. Box 57, Ituri, Congo; Michel.MandroNdahura@student.uantwerpen.be

**Keywords:** epilepsy, onchocerciasis, OV16 IgG, ivermectin, age

## Abstract

There is growing epidemiological evidence that onchocerciasis may induce epilepsy. High prevalence of onchocerciasis has been reported in onchocerciasis-meso and hyper-endemic regions. We aimed to determine the OV16 antibody prevalence in persons with epilepsy (PWE) in four onchocerciasis-endemic regions. PWE were identified during studies in Mahenge area (Tanzania), Kitgum and Pader districts (Uganda), the Mbam and Sanaga river valleys (Cameroon), and the Logo health zone (Democratic Republic of Congo). Exposure to *Onchocerca volvulus* was assessed by testing PWE for OV16 IgG4 antibodies using a rapid diagnostic test. The OV16 seroprevalence among PWE in the four onchocerciasis-endemic study sites ranged from 35.2% to 59.7%. OV16 seroprevalence increased with age until the age of 39 years, after which it decreased drastically. Our study suggests that, in onchocerciasis-endemic regions, epilepsy in young people is often associated with onchocerciasis, while epilepsy in older persons seems unrelated to *O. volvulus* exposure.

## 1. Background

Onchocerciasis elimination programs using community directed treatment with ivermectin (CDTI) have significantly reduced *Onchocerca volvulus* (*O. volvulus*) transmission and onchocerciasis associated blindness (river blindness) in many African countries [1]. Nevertheless, despite many years of CDTI, there is still active onchocerciasis transmission in many areas [2]. The explanation for this could be the absence or sub-optimal performance of onchocerciasis elimination programs with low ivermectin uptake in certain areas.

Currently, there is very strong epidemiological evidence that onchocerciasis is able to induce epilepsy [3,4,5], including the nodding syndrome in previously healthy children 3–18 years old [6]. This type of epilepsy is called onchocerciasis-associated epilepsy (OAE) [6]. OAE is a major public health problem in onchocerciasis-endemic areas, where onchocerciasis elimination programs are working sub-optimally or where such programs still need to be implemented [3,7] but disappear when onchocerciasis is eliminated from the area [5,8]. In recent years, we conducted many epidemiological studies to investigate the association between onchocerciasis and epilepsy in areas with high ongoing or past onchocerciasis transmission [9]. During these studies, persons with epilepsy (PWE) were tested for the presence of *O. volvulus* OV16 antibodies. In this paper, we describe the OV16 seroprevalence in PWE, taking into account their age, gender, and past ivermectin use.

## 2. Methods

Ethical approval was obtained from the Ethics committee of the School of Public Health in Kinshasa, (Logo: January 2017, ESP/CE/006/2017), the ethical committee of the National Institute for Medical Research, Tanzania (NIMR/HQ/R.8a/Vol.IX/2278) the Ethics committee of the Makerere University and St. Mary’s Hospital Lacor, Uganda (LHIREC 001/02/2017), the National ethical committee for the public health research, Cameroon (No: 2017/02/875/CE/CNERSH/SP), and the Ethics Committee of the Antwerp University Hospital (24 May 2017, B300201733011). Informed consent was obtained from all study participants.

Most PWE were identified during door-to-door epilepsy prevalence surveys in onchocerciasis-endemic areas with different histories of CDTI implementation. In Mahenge, Ulanga district, Tanzania [10,11], CDTI had been implemented for more than 20 years, Kitgum and Pader districts, Uganda [5], CDTI had been implemented since 2009, and in Cameroon, in Bilomo (Mbam valley), mass drug administration of ivermectin was implemented in 1998 and around mid-1990s in Kelleng (Sanaga valley) [12]. Other PWE were enrolled during the screening of PWE for participation in a clinical trial to investigate the effect of ivermectin on the frequency of seizures in persons with *O. volvulus* infection in the Logo health zone, Ituri province, Democratic Republic of Congo (DRC) [13]. In the Logo health zone, CDTI had never been implemented [14].

Recruitment of PWE was done following a two-step approach: screening using a validated questionnaire [15], followed by epilepsy case confirmation by a neurologist or clinician trained in epilepsy. This questionnaire was translated and adapted to make the questions more understandable by the local population [16]. The socio-demographic information of PWE was obtained, as well as their ivermectin use during the last CDTI round. Consented participants were finger-pricked, and a few drops of blood was collected for the detection of OV16 antibodies using a rapid diagnostic test (SD Bioline Onchocerciasis IgG4 rapid test, Abbott Standard Diagnostics, Inc., Yongin, Korea). The testing procedures were aseptic and followed the manufacturer’s instructions.

## 3. Statistical Analysis

The collected data and OV16 test results (positive or negative) were entered into electronic spreadsheets and prepared for analysis. Continuous variables were summarized using median and interquartile range (IQR), while proportions were expressed as counts and percentages. The Cochran–Armitage trend test [17] was used to assess OV16 seroprevalence trends across age groups. A logistic regression model was used to assess the association between age and OV16 seroprevalence adjusted for gender and study sites and past ivermectin exposure. The model goodness-of-fit was assessed using deviance and the Pearson chi-square test. Data were analyzed using SAS 9.4 (SAS Institute Inc., Cary, NC, USA) and R version 4.0.2, and a two-sided 5% significance level was used.

## 4. Results

Overall, OV16 serologic data of 760 PWE were analyzed. The OV16 seroprevalence among PWE in the four onchocerciasis-endemic study sites ranged from 35.2 to 59.7% (Table 1).

OV16 seroprevalence increased with increasing age in the younger age-groups but progressively decreased among older PWE (Cochran–Armitage trend test statistics of −4.75 and a two-sided *p*-value of <0.001) (Table 2).

The multivariable analysis revealed a significant association between age and OV16 serostatus (Table 3). Sex did not influence this association (Appendix A). Among younger participants (up to 39 years), increasing age was associated with increasing odds for OV16 seropositivity; conversely, in the older PWE (above 39 years), the probability of OV16 seropositivity decreased with increasing age (Figure 1).

## 5. Discussion

The increase of OV16 seroprevalence with increasing age up to the age of 39 years was expected among the enrolled PWE, as the same trend was also observed in the general population in onchocerciasis-endemic regions [18,19,20]. However, the decrease in OV16 prevalence in the PWE of 39 years and above was unexpected. Few population-based studies have investigated the OV16 antibody response in the older age groups and the results are not consistent. In a study conducted in an onchocerciasis-endemic area in Yemen, where ivermectin was never distributed, the OV16 seroprevalence continued to rise with age in men but decreased in women after the age of 39 years [20]. Differences according to the sex may be due to the fact that men and women get exposed to infected blackflies at different ages in their life because of different socio-economic activities, such as farming. Indeed, both younger and older men in Yemen are likely to work daily in the fields where they can be bitten by blackfly vectors, but young women tend to spend more time at the river side (for example, to wash clothes) than older women, who tend to remain in the houses, away from infective blackfly bites [20].

In a study conducted in Togo after decades of *Simulium damnosum* vector control and mass drug administration of ivermectin, the mean OV16-specific IgG4 reactivity was often low during the first 2 decades of life; from 16 years onwards, an enhanced responsiveness was observed and from 20 years and older, the mean participants’ serologic IgG4 responses to Ov16 continued to rise steadily until the fifth decade [18]. In another study in Togo, the OV16 prevalence increased up to the age of 41 and then remained at nearly the same level [19]. A previous study in the Logo health zone (DRC) found that the proportion of positive OV16 test results among the general population increased with increasing age, up to the age of 39 years. After 39 years, there was a slight decrease but the seroprevalence remained high [21]. In that same study, OV16 tests were positive in 18 (58.1%) of 31 participants who had previously received ivermectin and in 260 (29.5%) of 881 participants who had never been treated with ivermectin. Twenty-five (46.3%) of the 54 persons with epilepsy on whom OV16 tests were performed were positive, compared to 253 (29.5%) of 858 subjects without epilepsy (*p* = 0.014) [21], suggesting that onchocerciasis was more prevalent among PWE.

The drastic decrease of the OV16 seroprevalence among PWE in our study after the age of 39 years is in contrast with the observed OV16 seroprevalence among the general population in the DRC [21], Yemen [20], and Togo [18]. The most likely explanation for this observation is that the epilepsy beyond the age of 39 years in our study population was unrelated to onchocerciasis. Most of the younger PWE in the study areas met the criteria of onchocerciasis associated epilepsy (OAE) [10,12,22,23]. OAE typically develops in onchocerciasis-exposed children between the ages of 8 and 12 and has a high mortality, with very few cases reaching the age of 30 years [24]. It is therefore plausible that the older PWE with negative OV16 test results in our study did not develop OAE during their childhood/adolescence, but rather experienced epilepsy onset during the later decades of their lives. Frequent causes of epilepsy at older age include cerebrovascular accidents, head trauma, cerebral infections, or tumors [25,26] and those conditions are unrelated to onchocerciasis. Moreover, in all study sites, except the Logo health zone (DRC), older PWE had been exposed to many years of ivermectin and this may have decreased the prevalence of OV16 seropositivity [27]. In addition, PWE generally avoid coming close to the rivers because of the risk of drowning during seizures and therefore are less exposed to infected blackflies. As a consequence, after many years of non-exposure, OV16 antibodies may have disappeared [27]. The level *O. volvulus* microfilarial load in children was shown to be a predictor for developing OAE later in life [3,4]; however, the pathophysiological mechanism on how the *O. volvulus* parasite causes the seizures remains unknown. No parasites or *O. volvulus* DNA and no *Wolbachia* DNA (an *O. volvulus* endosymbiont) were found in cerebrospinal fluid [28] or in brain tissue of persons who had died of OAE [29]. An auto-immune mechanism induced by neurotoxic *O. volvulus* cross-reacting antibodies was proposed [30], but this hypothesis still needs to be confirmed.

Our findings should be interpreted in light of a number of limitations, including the fact that the data on OV16 seroprevalence among the general population in our study sites was available only for the Logo health zone (DRC). Another limitation is that no other laboratory test or brain imaging investigations were done to identify the epilepsy etiology in our study participants. Moreover, the presence of *O. volvulus* antibodies was only assessed using an OV16 rapid diagnostic test, which is less sensitive compared to OV16 ELISA [31].

In conclusion, our study confirms that OV16 seroprevalence is high among PWE, particularly in the younger age groups. Our findings suggest that, in onchocerciasis-endemic areas, epilepsy in young people may be associated with exposure to *O. volvulus* and that, in older PWE, additional investigations need to be performed to identify risk factors for epilepsy other than onchocerciasis.

## Figures and Tables

**Figure 1 pathogens-09-00847-f001:**
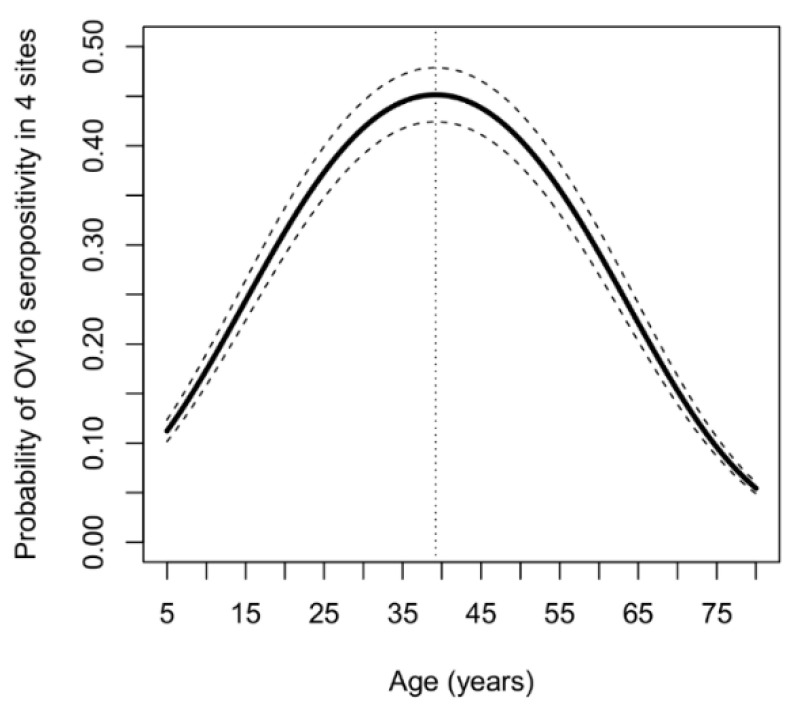
The probability OV16 seropositivity as a function of age, taking into account gender and ivermectin history in four onchocerciasis-endemic areas. Solid black lines represent the point estimates and dashed lines indicate pointwise 95% Wald-based confidence bands.

**Table 1 pathogens-09-00847-t001:** Characteristics of persons with epilepsy in the four study sites.

Characteristics	Logo, DRC (*n* = 391)	Mbam and Sanaga Valley, Cameroon (*n* = 77)	Mahenge, Tanzania (*n* = 187)	Kitgum and Pader, Uganda (*n* = 105)
Age in years: median (IQR)	20.0 (14.0–29.0)	26.0 (21.0–31.0)	24.0 (18.0–34.5)	20.0 (16.0–24.0)
Male gender: *n* (%)	205 (52.4)	34 (44.2)	85 (47.2)	56 (53.3)
Past ivermectin use: *n* (%)	0	51 (66.2)	89 (47.6)	82 (78.1)
OV16 positive: *n* (%)	149 (38.1)	46 (59.7)	102 (54.5)	37 (35.2)

Interquartile range (IQR), Democratic Republic of Congo (DRC).

**Table 2 pathogens-09-00847-t002:** Age-specific OV16 seroprevalence.

Age Group	OV16 Seroprevalence, *n* (%)
10 years or younger (*n* = 79)	7 (8.9)
11–20 years (*n* = 253)	103 (40.7)
21–30 years (*n* = 235)	128 (54.5)
31–40 years (*n* = 93)	39 (51.6)
41–50 years (*n* = 39)	17 (53.8)
51 years or older (*n* = 47)	15 (46.8)

**Table 3 pathogens-09-00847-t003:** Predictors of a positive OV16 test among persons with epilepsy in four onchocerciasis endemic areas.

Parameter	Coeff	95% CI		*p* Value
Age (years)	0.1256	0.0780	0.1733	<0.0001
Age*age	−0.0016	−0.0023	−0.0009	<0.0001
Female gender	0.0994	−0.2225	0.4213	0.5451
Ivermectin intake vs no intake during last CDTI round	−0.3471	−0.9851	0.2908	0.2862
Country site (CMR vs. DRC)	0.384	−0.2788	1.0468	0.2562
Country site (TZD vs. DRC)	0.3682	−0.3455	1.0819	0.3120
Country site (UG vs. DRC)	−0.3582	−1.086	0.3696	0.3347

Estimated coefficient from logistic regression (Coeff); Wald confidence interval (CI); Mbam and Sanaga valley, Cameroon (CMR); Logo, Democratic Republic of Congo (DRC); Mahenge, Tanzania (TZD); Kitgum and Pader, Uganda (UG).

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
