# Peer review of "OV16 Seroprevalence among Persons with Epilepsy in Onchocerciasis Endemic Regions: A Multi-Country Study"

_pathogens, 2020, doi:10.3390/pathogens9100847_

Round 1
Reviewer 1 Report
Overall a well written and interesting paper, on PWE and O. volvulus prevalence , which is an important emerging neglected topic. However, I have a number of comments regarding the paper which need to be addressed.
Background - 1. this is very short and more in line with a 'short communication' rather than a full article. So I wonder if it should be presented as such, if not it may be helpful to provide an additonal paragraph on oncho elimination / ivermectin use MDA and the OV16 test to give a broader perspective.
Methods - 1. more information on the locations (sub-district level) and if they were CDTi area would be helpful. 2. There is no comparison with people without epilepsy which would have been a much stronger study and something the researchers should consider in the future as we don't know if the levels in the PWE are higher, similar or lower than the communities in which they live. Please not this as a limitation/ future opportunity in discussion 3. There are limitations with OV16 rapid test and these need to be noted somewhere (in methods or discussion). 4. the aim is to look at PWE taking age, gender, MDA history into account but the gender and ivermectin are not note in methods section - 5. Having the 'validated questionnaire in English as a supplmentary fiel woudl be a good contribution - reference 12 indicated this is from a paper in 2006 potentially in French and not an open access journal which would make it hard for people to undertand how this survey was done - I am curious myself as I think others would be.
Results - 1. I do not think that there difference in Table 2 between 53.8% in the 41-50 year old is the signicant to the 51+ years 46.8% 2. Was this age pattern the same across all communities and by gender? That may be a more informative table. Often in filariasis related infections, the male vs female age patterns are different. 3. how was the ivemectin expsure measure - where people asked about number of rounds - please describe in methods so it is clear what the results mean
Discussion - the limitations need to be expanded to put into perspective the lack of a control group and use of only one diagnostic test
Reviewer 2 Report
Dear Authors: Thank you very much for your paper.
You addressed the important association between onchocerciasis and epilepsy. Can you address or hypothesize what mechanisms would lead to epilepsy in patients with OAE and nodding syndrome?
How do you explain the high prevalence of epilepsy despite 20-yrs of CDTI. If this remains high what would be the importance of doing this test?. Mmbando High prevalence of epilepsy in two rural onchocerciasis endemic villages in the Mahenge area, Tanzania, after 20 years of community directed treatment with ivermectin. Infect Dis Poverty. 2018 Jun 20;7(1):64.
Could the presence of the presence of epilepsy and onchocerciasus may just be association and not causation, and its prevalence is just incidental ?
Can you add to your references the report of Low incidence of epilepsy and low prevalence of onchocerciasis? Siewe JNF, Colebunders R. Low prevalence of epilepsy and onchocerciasis aft more than 20 years of ivermectin treatment in the Imo River Basin in Nigeria. Infect Dis Poverty. 2019 Jan 23;8(1):8er
Round 2
Reviewer 1 Report
The revisions are acceptable